# Elucidation of Novel Therapeutic Targets for Breast Cancer with *ESR1-CCDC170* Fusion

**DOI:** 10.3390/jcm10040582

**Published:** 2021-02-04

**Authors:** Jae Heon Jeong, Jae Won Yun, Ha Young Kim, Chan Yeong Heo, Sejoon Lee

**Affiliations:** 1Integrated Major in Innovative Medical Science, College of Medicine, Seoul National University, Seoul 08826, Korea; jheonii@snu.ac.kr; 2Interdisciplinary Program for Bioengineering, College of Engineering, Seoul National University, Seoul 08826, Korea; hkim247@snu.ac.kr; 3Department of Plastic and Reconstructive Surgery, Seoul National University Bundang Hospital, Seongnam 13620, Korea; 4Veterans Medical Research Institute, Veterans Health Service Medical Center, Seoul 08826, Korea; jwyunmd@gmail.com; 5Department of Plastic and Reconstructive Surgery, College of Medicine, Seoul National University, Seoul 03080, Korea; 6Precision Medicine Center, Seoul National University Bundang Hospital, Seongnam 13620, Korea; 7Department of Pathology and Translational Medicine, Seoul National University Bundang Hospital, Seongnam 13620, Korea

**Keywords:** ESR1, CCDC170, breast cancer, gene fusion, TCGA, bioinformatics, drug repositioning

## Abstract

Among the various types of breast cancer, the luminal B subtype is the most common in young women, and *ESR1-CCDC170* (E:C) fusion is the most frequent oncogenic fusion driver of the luminal B subtype. Nevertheless, treatments targeting E:C fusion has not been well established yet. Hence, the aim of this study is to investigate potential therapies targeting E:C fusion based on systematic bioinformatical analysis of the Cancer Genome Atlas (TCGA) data. One thousand related genes were extracted using transcriptome analysis, and major signaling pathways associated with breast cancer were identified with over-representation analysis. Then, we conducted drug-target network analysis based on the OncoKB and CIViC databases, and finally selected potentially applicable drug candidates. Six major cancer-related signaling pathways (p53, ATR/ATM, FOXM1, hedgehog, cell cycle, and Aurora B) were significantly altered in E:C fusion-positive cases of breast cancer. Further investigation revealed that nine genes (*AURKB*, *HDAC2*, *PLK1*, *CENPA*, *CHEK1*, *CHEK2*, *RB1*, *CCNA2*, and *MDM2*) in coordination with E:C fusion were found to be common denominators in three or more of these pathways, thereby making them promising gene biomarkers for target therapy. Among the 21 putative actionable drugs inferred by drug-target network analysis, palbociclib, alpelisib, ribociclib, dexamethasone, checkpoint kinase inhibitor AXD 7762, irinotecan, milademetan tosylate, R05045337, cisplatin, prexasertib, and olaparib were considered promising drug candidates targeting genes involved in at least two E:C fusion-related pathways.

## 1. Introduction

Breast cancer, aside from skin cancer, is the most commonly diagnosed cancer in women worldwide [1]. Recent statistics report the emergence of 250,000 new cases of breast cancer solely in 2017, contributing to the 12% of women diagnosed with breast cancer in the United States [2]. Molecular classification divides breast cancer into four major classes: luminal A, luminal B, human epidermal growth factor receptor 2 (HER2)-enriched (HER2-E), and basal-like subtype [2]. Among them, luminal B remains to be the most common subtype in young women, accounting for 15%–20% of the total breast cancer cases, and within luminal B, *ESR1-CCDC170* fusion-positive subtype, constituting 6% to 8% of the luminal B class, persists to be the most dominant subtype [3,4,5,6,7,8,9].

*ESR1-CCDC170* fusion causing chimeric mRNA is known to be formed by a tandem duplication at the 6q25.1 location on a coiled-coil domain containing 170 (*CCDC170*) adjacent to the *ESR1* gene [8,10]. It has been reported that the polymorphism of the *CCDC170* gene correlates with breast cancer susceptibility [11,12]. *ESR1-CCDC170* fusion-positive cancers treated with endocrine therapy showed reduced treatment efficiency in mouse models [13]. Although its effect has been studied in relation to ovarian cancer, the molecular signaling involved in the induction of *ESR1-CCDC170* fusion-positive breast cancer has yet to be elucidated [14].

Here, we systematically analyzed the molecular pathological features of *ESR1-CCDC170* fusion-positive breast cancer through the data analysis of the Cancer Genome Atlas (TCGA) and identified the activated oncogenic pathways. In addition, putative target genes and actionable drugs were inferred and prioritized by performing network analysis using both transcriptomic signatures and drug-target databases, such as OncoKB and CIViC.

## 2. Experimental Section

### 2.1. Sample Acquisition and Quality Control

Gene level 3 (RNA-seq by expectation-maximization, RSEM) mRNA expression with normalized read count values of TCGA breast cancer carcinoma (BRCA) was obtained from the Broad GDAC Firehose website (https://gdac.broadinstitute.org). Related clinical feature data, including information about the samples’ mutation annotation format (MAF) files, molecular subtypes, and tumor-node-metastasis (TNM) stages, were obtained from the website mentioned above.

### 2.2. Case-Control Selection

A previous study confirmed 319 fusion genes in TCGA clinical breast cancer tumors [15]. Unlike other in-frame fusion genes, *ESR1-CCDC170* is known as a breast cancer-specific oncogenic fusion gene. Using the TCGA fusion gene data portal (Jackson Laboratory, https://www.tumorfusions.org), we identified 11 samples of *CCDC170* fusion, which were cross-checked with increased *CCDC170* expression level. Furthermore, only tumor samples that had the barcode 01A (primary solid tumor) were selectively chosen by disregarding other types of tumor samples, 11A (normal) or 06A (metastasized). Among the remaining samples, 50 samples with the highest expression of *CCDC170* were confirmed as upregulated controls for analyzing the network within the noncoding region of the fusion gene. From the upregulated control samples, 2 outlier samples were filtered out using the IQR (interquartile range) method. The same number of controls (*n* = 48) with the lowest expression of *CCDC170* were then selected from the remaining samples.

### 2.3. Selection of Genes Affected by ESR1-CCDC170 Fusion

RNA expression data from TCGA were made into a two-dimensional matrix composed of the selected 11 fusion samples and 48 control samples. Each column represents the patient ID, while each row represents the gene name. Based on the RNA expression matrix, variance tests were conducted using independent two-sample *t*-tests. To select genes in coordination with *CCDC170* in the RNA expression, *t*-tests were performed between E:C fusion-positive and fusion-negative cases. Mostly affected 1000 genes were selected (adjust *p*-value < 2.0 × 10^−8^).

### 2.4. Pathway Analysis via ConsensusPathDB (CPDB) and Over-Representation

The selected 1000 genes that correlated to the reference gene (*CCDC170*) from the aforementioned RNA expression data were used to perform over-representation analysis (ORA) via ConsensusPathDB (CPDB, http://cpdb.molgen.mpg.de/CPDB 11th September 2020). We inputted a gene list with the option of Entrez Gene using pathway-based sets with a minimum overlap input list (*n* = 2) and *p*-value cutoff (*p*-value < 0.01). A total of 113 biological pathways were merged and curated by CPDB from the following sources, according to data from BioCarta (https://maayanlab.cloud/Harmonizome/dataset/Biocarta+Pathways), INOH [16], KEGG [17], NetPath [18], PID [19], Reactome [20], and WikiPathways [21]. In consideration of the ontological characteristics and the proportion of duplicated genes, the pathways, which were enriched with selected 1000 genes (*q*-value < 0.05), were condensed into 15 cancer-related pathways, and their components were 184 genes.

### 2.5. Druggable Pathway Analysis via CIViC and OncoKB

The “Clinical Evidence Summaries” data, released on 1 October 2017, were downloaded from the Clinical Interpretations of Variants in Cancer (CIViC) website (https://civic.genome.wustl.edu/releases), and the “Actionable Variants” data were accessed and downloaded on 17 October 2017 from the OncoKB website (http://oncokb.org/). A total of 673 CIViC variants (181 genes) with expected therapy efficacy in 148 OncoKB actionable variants (53 genes) were integrated. A total of 113 *CCDC170*-correlated genes were matched to the CIViC and OncoKB variants.

### 2.6. Statistical Analysis and Data Visualization

The open software R version 3.4.3 was used to process all statistical analyses for selecting genes correlated to *CCDC170*, including the variance test and independent two-sample *t*-test. An RNA expression heatmap was also visualized using ComplexHeatmap, a package for R. A KEGG mapper (https://www.genome.jp/kegg/mapper.html 11th September 2020) was used to visualize target pathways related to DNA damage response. Cytoscape version 3.5.3 was used to analyze and express the complex network between targetable drugs and therapeutic agents. Our study defined statistical significance with a *p*-value of < 0.05 and false detection rate (FDR) with a *q*-value of <0.001.

## 3. Results

### 3.1. Clinico-Pathological Characteristics

We checked the clinico-pathological characteristics of 11 *ESR1-CCDC170* fusion-positive and 48 fusion-negative patients among 1095 breast cancer patients in Broad GDAC Firehose (Figure 1, Table 1). Two significant differences were identified between fusion-positive and negative patients.

First, *CCDC170* fusion-positive patients had a high rate of ER-positive (90.9%) and PR-positive (63.6%), whereas fusion-negative patients displayed significantly lower rates of 20.8% and 4.3%, respectively (*p* < 0.05). Additionally, HER2 immunohistochemistry (IHC) results showed a significantly higher rate of 3+ for fusion-positive patients than for patients with fusion-negative (44.4% vs. 9.4%, *p* < 0.05, Table 1). According to the findings above, *CCDC170* fusion-positive BRCA appears to closely resemble characteristics typical of triple-positive breast cancer in this cohort.

Second, the pathological subtype of fusion-positive patients was found to have a high proportion of the luminal A (45.5%) and luminal B subtypes (45.5%), while 90.9% of the 48 cases with the lowest *CCDC170* expression were found to be basal type (*p* < 0.05, Table 1). Taken together, the *CCDC170* fusion-positive BRCA showed a mutually exclusive relationship with the basal-type breast cancer cells.

On the other hand, no significant differences in age, sex, vital status, and TNM stage were observed between the two groups. In addition, the five gene variants (*PIK3CA*, *CDH1*, *TP53*, *BRCA1*, and *BRCA2*) frequently found in BRCA showed no significant difference between the two groups. Based on these results, *CCDC170* fusion-positive BRCA patients have distinct pathological characteristics in terms of tumor subtype and triple positive tendency.

### 3.2. Key Pathways and Genes Altered in ESR1-CCDC170 Fusion-Positive Breast Cancer

One thousand genes were obtained by an independent *t*-test (*q* < 2.0 × 10^−8^) and inputted for performing the over-representation analysis (ORA) of the ConsensusPathDB website to select cancer-related pathways. As a result, a total of six cancer-related pathways (p53, ATR/ATM, FOXM1, hedgehog, Cell cycle, and Aurora B-related signaling pathways) were discerned.

In the six major cancer-related pathways, 137 genes were significantly over- or under-expressed in the *CCDC170* fusion-positive cases compared with the *CCDC170* fusion-negative controls (Figure 2, Appendix A).

Of the six pathways, two concerning p53- and ATR/ATM-related signaling pathways were associated with DNA damage response. Mapping with the KEGG pathway revealed 22 genes that are involved in the p53-related pathway and 17 genes in the ATR/ATM-related pathway. Both pathways are highly relevant to the promotion and maintenance of the cell cycle (Figure 3). Genes with multiple hits of more than two that coincide for both p53- and ATR/ATM-related signaling pathways are *CCNA2(CycA)*, *MDM2*, *CHEK1(Chk1)*, and *CHEK2(Chk2)*.

There were 39 genes involved in multiple pathways (Figure 4, Appendix A), of which *AURKB*, *HDAC2*, *PLK1*, *CENPA*, *CHEK1*, *CHEK2*, *RB1*, CCNA2, and *MDM2* were included in at least three pathways that are important for tumor proliferation and maintenance specific to *ESR1-CCDC170* fusion-positive BRCA patients.

Further investigation of the 48 samples with the highest mRNA levels of *CCDC170* with the differentially expressed gene (DEG) analysis showed similar patterns as the TCGA data obtained above in fusion-positive samples when compared with the control samples (Appendix A, Appendix A). This suggests that a similar cell signaling is activated not just with fusion but with other possibilities in *CCDC170* over-expression.

### 3.3. Identification of Actionable Targets and Potential Therapeutic Choice Using Network Analysis

Actionable target genes and potentially available drugs were extracted by inputting the 137 genes in the following drug databases: CIViC (*n* = 673) and OncoKB (*n* = 262). *ESR1*, *CDK4*, *RAD50*, *CHEK1*, *MDM2*, and *SMO* were mapped as targetable genes. Results indicated the following drug-target relationships: letrozole, palbociclib, fulvestrant, AZD9496, and tamoxifen for *ESR1*; palbociclib, alpelisib, ribociclib, and dexamethasone for *CDK4*; checkpoint kinase inhibitor AZD7762 and irinotecan for *RAD50*; cisplatin, prexasertib, and olaparib for *CHEK1*; milademetan tosylate and RO5045337 for *MDM2*; and PSI, vismodegib, patidegib, and arsenic trioxide for *SMO* (Figure 5).

Observing the druggable target genes associated with the main pathways of E:C fusion-positive BRCA, *ESR1* and *CDK4* genes were included in the FOXM1-related signaling pathway; *RAD50* in the ATR/ATM-related signaling pathway; *CHEK1* and *MDM2* genes in the P53-related signaling pathway; *CDK4*, *RAD50*, and *MDM2* genes in the cell cycle-related signaling pathway; and *SMO* genes in the hedgehog-related signaling pathway. Interestingly, four of the six targetable genes, *CDK4*, *RAD50*, *CHEK1*, and *MDM2*, were involved in two or more major cancer-related pathways. In the case of *MDM2*, three of the six pathways associated with E:C fusion-positive were identified to be involved.

## 4. Discussion

In this study, the characteristics of an ER-positive molecular subtype in *CCDC170*-subtype breast cancer were identified, and genes specifically regulated in E:C BRCA were identified and screened for BRCA-related signaling pathways. Additionally, information regarding optimal treatment targets and drugs for targeted therapy was provided. E:C fusion-positive BRCA requires a new therapeutic approach to overcome its relatively low response to hormone therapy [6,13,22,23,24].

A recent study on a potential targeted therapy for E:C BRCA performed by Li et al. was met with limitations with regard to a restricted number of cell line samples and proteins [13]. Our study has addressed this issue by performing analysis on a sufficient number of case-control TCGA human cancer samples and systematically testing the DEGs using more than 20,000 genes and cancer-specific pathways. Finally, we were able to propose a number of potential drugs with promising therapeutic effects.

The common early treatment options for breast cancer are generally divided into conventional chemotherapy (Adriamycin, cyclophosphamide, paclitaxel, and docetaxel), endocrine therapy (tamoxifen, letrozole, anastrozole, and exemestane), ERBB-targeted therapy (trastuzumab and pertuzumab), and combination treatment methods according to the pathological and molecular classification of breast cancer [2]. In the case of metastatic breast cancer, CDK4/6 inhibitor and PARP inhibitor are considered to be additional options [2]. On the other hand, many resistance mechanisms for drug therapy in breast cancer have been reported as follows: loss of estrogen receptor, deregulation of cell cycle for endocrine therapy, incomplete blockade of HER receptors, activation of the PI3K pathway, over-expression of estrogen receptor for HER2 inhibitors, polyclonal RB1 mutations for CDK 4/6 inhibitors, and so on [25,26]. Hence, finding novel therapeutic strategies using drug repositioning analysis is crucial for modern breast cancer treatment.

Among the repositioned drugs inferred in our study, CDK4/6 inhibitor (palbociclib), cisplatin, and PARP inhibitor are the drugs used as standard treatments for breast cancer patients with or without metastasis. On the other hand, AZD9496, alpelisib, dexamethasone, checkpoint kinase inhibitor AZD7762, irinotecan, cisplatin, prexasertib, milademetan tosylate, R05045337, PSI, vismodegib, patidegib, and arsenic trioxide are seen as putative actionable drugs that can be used for E:C fusion-positive BRCA proceeding in vitro and in vivo validation.

*AURKB*, *HDAC2*, *PLK1*, *CENPA*, *CHEK1*, *CHEK2*, *RB1*, and *MDM2* genes, which were included in at least three pathways, are expected to play an important role in the promotion and maintenance of *CCDC170*-subtype breast cancer. For instance, *PLK1* may act as a tumor suppressor gene that regulates estrogen receptor (ER)-regulated gene transcription in breast cancer [27]; *RB1* gene, also a tumor suppressor gene, however, is frequently lost in triple-negative breast cancer [28]; *CENPA* is a significant prognostic marker for ER-positive patients [29]; and *HDAC2* and *CHEK2* genes have been significantly correlated to *CCDC170* fusion subtype and have been reported to be associated with DDR functioning [30,31], which is also suggestive of the *CCDC170* fusion subtype’s relationship with DDR.

In addition, we investigated whether there is a biological difference between the E:C fusion-positive group and the CCDC170 high-expression group without fusion. We found that there showed no major difference in cancer signaling except in several minor pathways, including the cilium assembly pathway and integrins in the angiogenesis pathway (Appendix A).

In summary, this study presents core biomarkers and potentially actionable drugs specific to E:C fusion-positive breast cancer. Via in vitro experimentation, these candidates were confirmed to be strongly associated with this type of cancer, and their roles were verified by discerning their associated signaling pathways. We hope that our findings will be the steppingstone for future investigations, leading to the promotion of a targeted cancer therapy.

## Figures and Tables

**Figure 1 jcm-10-00582-f001:**
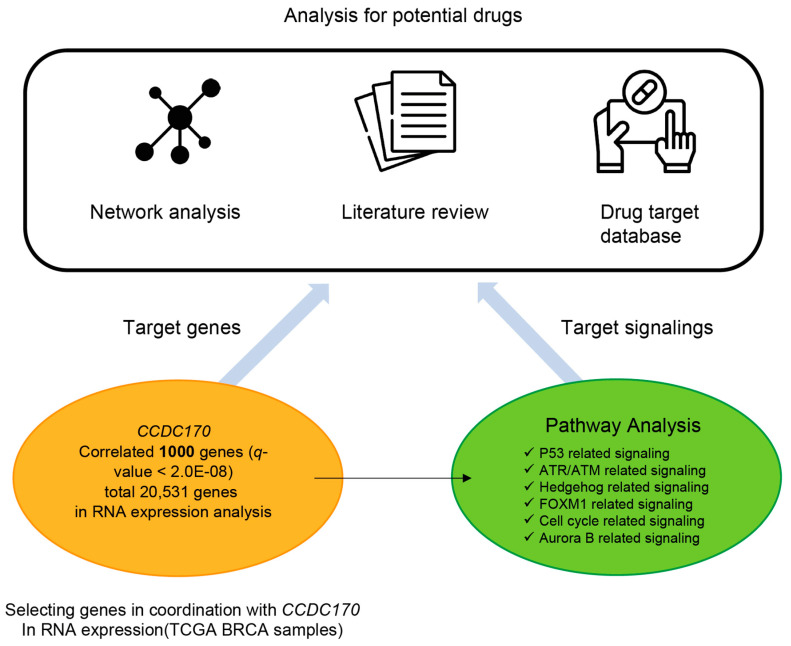
Overall schematics. Transcriptome data for breast cancer (BRCA) were obtained from the Broad GDAC Firehose database. Following the RNA measurement analysis of a total of 20,531 genes, 1000 genes correlated with *CCDC170* were selected (*q* < 2.0 × 10^−8^). Over-representation analysis of the 1000 genes demonstrated a significant relationship with six major cancer-related pathways (p53, ATR/ARM, hedgehog, FOXM1, cell cycle, and Aurora B). Potential gene targets and drug candidates were isolated via drug network analysis using a drug-target database on genes correlated to *CCDC170* and the literature review.

**Figure 2 jcm-10-00582-f002:**
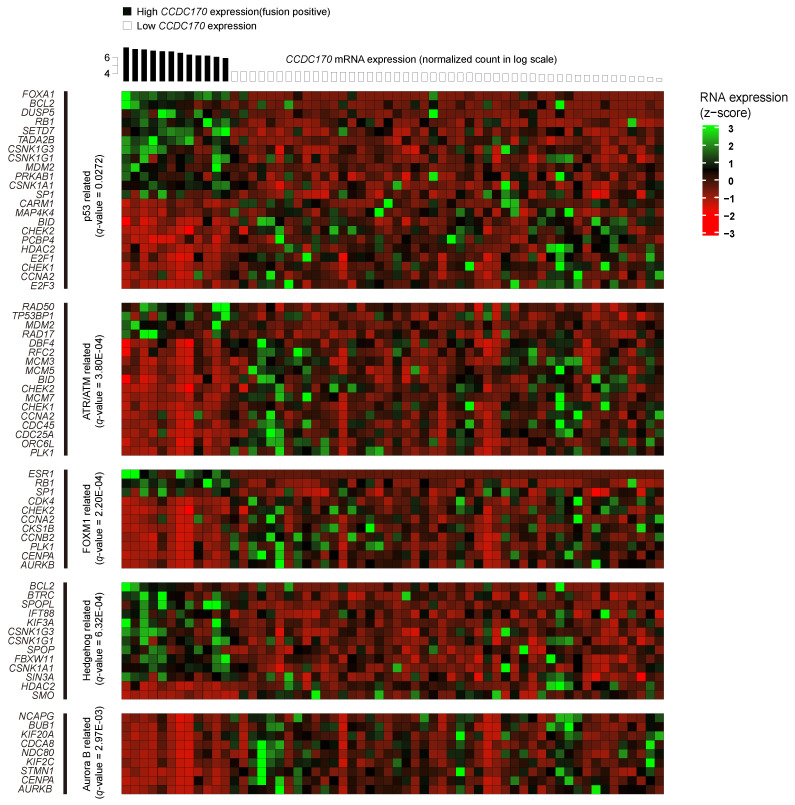
Gene expression heatmap of cancer-related pathways correlated with *CCDC170* RNA expression. Of the analyzed genes, 72 of the genes associated with P53, ATR/ATM, FOXM1, hedgehog, and Aurora B demonstrated significant differences in expression in *CCDC170* fusion-positive BRCA samples when compared with the control group. Over-representation analysis using ConsensusPathDB (CPDB) yielded statistically significant pathways related to cancer (*q* < 0.05). The *x*-axis is indicative of the sample, while the *y*-axis is indicative of its respective RNA expression. The RNA expression was converted into z-score prior to representation on the heatmap.

**Figure 3 jcm-10-00582-f003:**
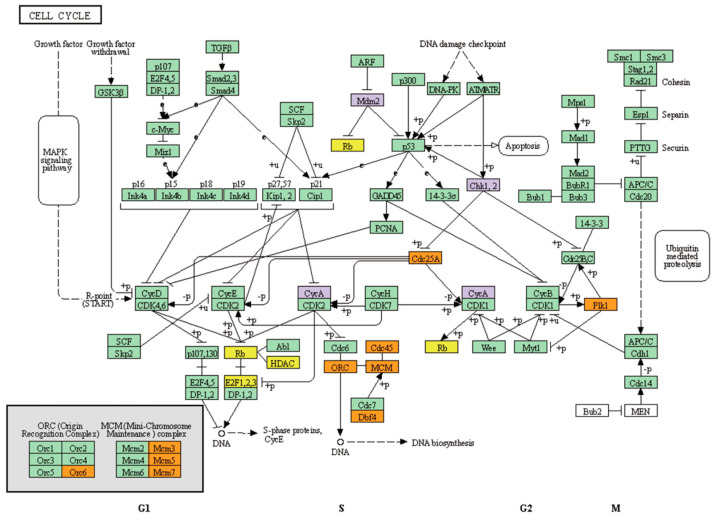
Over- and under-expressed genes are enriched in the human cell cycle pathway. The KEGG pathway map for the human cell cycle signaling pathway, has04110, was visualized using the KEGG mapper. Among the pathways, p53 and ATR/ATM shared a significant correlation with the identified genes. Genes associated with the p53 signaling pathway are boxed in yellow, ATR/ATM in orange, and common denominators for both pathways in purple.

**Figure 4 jcm-10-00582-f004:**
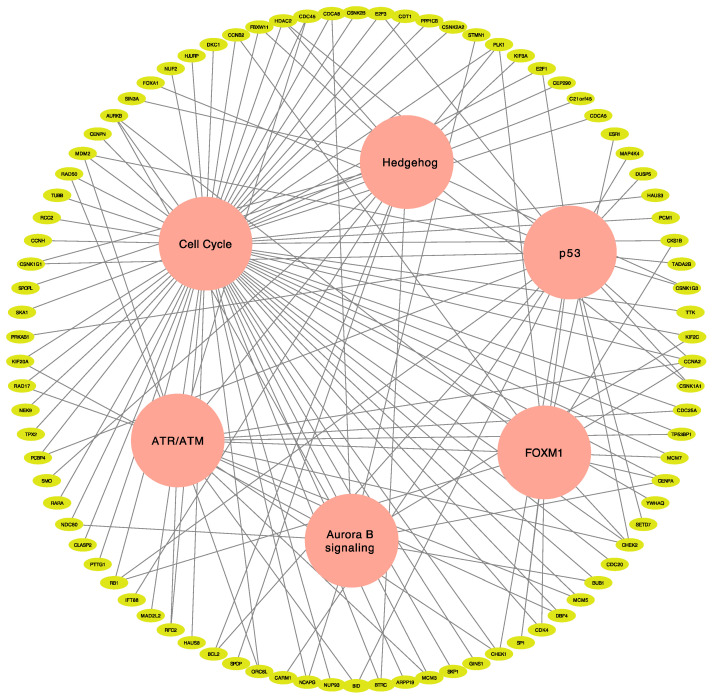
Putative target genes involved in multiple pathways of *ESR1-CCDC170* fusion-positive cancer. Six major cancer signaling pathways associated with p53, ATR/ATM, FOXM1, hedgehog, cell cycle, and Aurora B in accordance with their respective genes were visualized. Potential gene candidates involved in these pathways were discerned.

**Figure 5 jcm-10-00582-f005:**
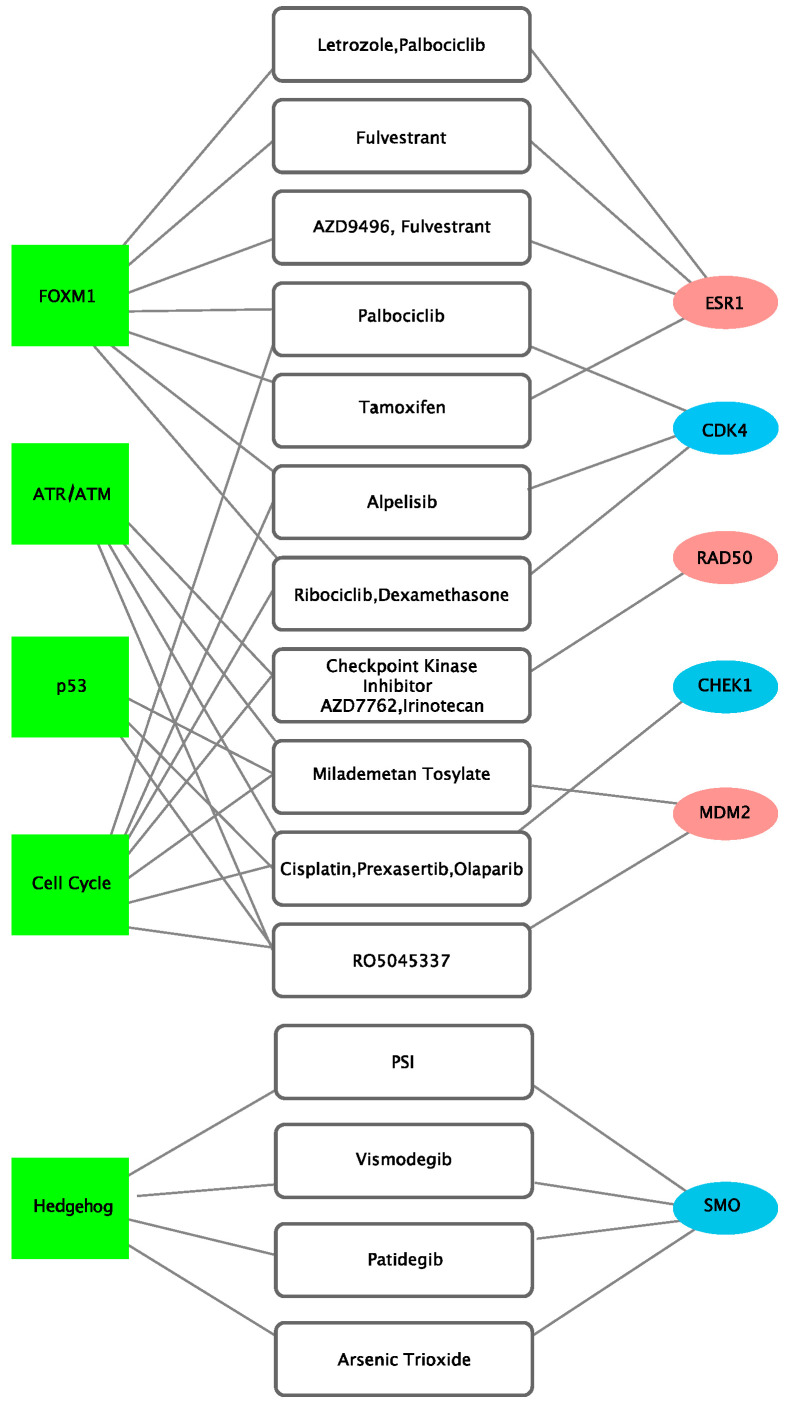
Drug-target network of *ESR1-CCDC170* fusion-positive BRCA cancer. Network visualization was demonstrated with Cytoscape, and drug-target relationship was identified with CIViC and OncoKB. Green boxes are representative of pathways, white boxes of drugs, and oval boxes of genes. Red oval boxes are genes that are over-expressed in fusion-positive cancer, whereas blue oval boxes are genes that are under-expressed in fusion-positive cancer.

**Table 1 jcm-10-00582-t001:** Comparisons in clinical and pathological characteristics of *ESR1-CCDC170* fusion-positive and negative BRCA patients and control cohorts. The clinical and pathological characteristics between *ESR1-CCDC170* fusion-positive, negative BRCA, and control cohorts were compared.

	Total (*n* = 1095)	Control (*n* = 48)	Fusion (*n* = 11)	*p*-Values
age	46–72	44–68	49–71	0.001
sex				
- female	1082 (98.8%)	48 (100.0%)	11 (100.0%)	
Vital status				1
- alive	991 (90.5%)	42 (87.5%)	10 (90.9%)	
- dead	104 (9.5%)	6 (13.6%)	1 (9.1%)	
stage				NS
- stage I	90 (8.3%)	2 (4.2%)	1 (9.1%)	
- stage Ia	85 (7.8%)	3 (6.2%)	1 (9.1%)	
- stage Ib	6 (0.6%)	0 (0.0%)	0 (0.0%)	
- stage II	6 (0.6%)	0 (0.0%)	0 (0.0%)	
- stage IIa	359 (33.0%)	24 (50.0%)	3 (27.3%)	
- stage IIb	257 (23.6%)	9 (18.8%)	3 (27.3%)	
- stage III	2 (0.2%)	0 (0.0%)	0 (0.0%)	
- stage IIIa	156 (14.4%)	4 (8.3%)	2 (18.2%)	
- stage IIIb	27 (2.5%)	0 (0.0%)	0 (0.0%)	
- stage IIIc	65 (6.0%)	3 (6.2%)	1 (9.1%)	
- stage Iv	20 (1.8%)	2 (4.2%)	0 (0.0%)	
- stage x	14 (1.3%)	1 (2.1%)	0 (0.0%)	
ER status				0
- positive	808 (77.2%)	10 (20.8%)	10 (90.9%)	
- negative	237 (22.5%)	38 (79.2%)	0 (0.0%)	
- indeterminate	2 (0.2%)	0 (0.0%)	1 (9.1%)	
PR status				0
- positive	700 (66.9%)	2 (4.3%)	7 (63.6%)	
- negative	342 (32.7%)	45 (95.7%)	4 (36.4%)	
- indeterminate	4 (0.4%)	0 (0.0%)	0 (0.0%)	
HER2 IHC				0
0	61 (9.8%)	7 (21.9%)	0 (0.0%)	
- 1+	270 (43.4%)	12 (37.5%)	4 (44.4%)	
- 2+	199 (32.1%)	10 (31.2%)	1 (11.1%)	
- 3+	90 (14.5%)	3 (9.4%)	4 (44.4%)	
subtype				0
- Basal	190 (18.0%)	42 (89.4%)	0 (0.0%)	
- Her2	82 (7.8%)	5 (10.6%)	1 (9.1%)	
- LumA	566 (53.6%)	0 (0.0%)	5 (45.5%)	
- LumB	217 (20.6%)	0 (0.0%)	5 (45.5%)	
*PIK3CA* mutation	301 (31.2%)	3 (6.8%)	1 (10.0%)	NS
*CDH1* mutation	96 (10.0%)	0 (0.0%)	1 (10.0%)	NS
*TP53* mutation	264 (27.4%)	29 (65.9%)	3 (30.0%)	NS
*BRCA1* mutation	11 (1.1%)	0 (0.0%)	0 (0.0%)	NS
*BRCA2* mutation	12 (1.2%)	1 (2.7%)	1 (10.0%)	NS

## Data Availability

The data is available in the following website: https://gdac.broadinstitute.org.

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
