# Peer review of "Elucidation of Novel Therapeutic Targets for Breast Cancer with ESR1-CCDC170 Fusion"

_jcm, 2021, doi:10.3390/jcm10040582_

Round 1

Reviewer 1 Report

In the study by Jae Heon Jeong et al., the characteristics concerning ESR1-CCDC170 fusion gene positive breast cancer were described by bio-informatic approach, more particularly genes specifically regulated, cancer related signaling pathways and potential targeted therapies for such breast cancer type. The results presented here provide new insights into the ESR1-CCDC170 fusion gene positive breast cancer molecular profile and could contribute to identified new therapeutic strategies but futher investigations (pré-clinical) are needed to clarifie and confirm the pathophysiological role of ESR1-CCDC170 fusion gene in breast cancer.

Moreover, there are some concerns.

Majors concerns:

Line  57-59 : Despite suspected, direct and robust clinical evidences of worst prognostic and therapeutic resistance associated with ESR1-CCDC170 fusion gene are still missing. Then authors should be more prudent and avoid misinterpretation of previous work (reference number 13 is exclusively preclinic).

The aim of the study (comparison between CCDC170 low and high expression level or decription of ESR1-CCDC170 fusion gene consequence?) and the selection of control samples are not clear. It seems that there are 4 groups in this study : the whole cohort (named "total" in table 1, whose exact case number remains unclear), the ESR1-CCDC170 fusion gene positive group (n=11), a positive control group displaying high CCDC170 mRNA expression without fusion gene (n=??) and a negative control group displaying low CCDC170 mRNA expression  (n= 48 or 44 ??). Selection of control groups (negative and positive) must be explained in details : cut-off to distinguish high and low expression level in comparison with the ESR1-CCDC170 fusion gene positive group, with 50, 48 or 44 cases ? does the total cohort correspond to ESR1-CCDC170 fusion positive group + positive and negative control groups + the remaining cases (normal or medium expression level of CCDC170 protein? Moreover a clear description of clinical characteristics of these 4 groups is expected.  In table 1, we don’t( know if the control group correspond to high or low CCDC170 expressing group ?). In addition, exclusion of metastatic cases should be explained (why ?). It seems that they are not fully excluded because in Table 1, stage Iv, which does not exist, is probably stage IV, which is the metastatic stage.

Authors have written about “39 genes identified in Figure 4”, but the list of these genes was not given and must be provided because figure 4 is not contributive to identified them.  

ESR1-CCDC170 fusion gene has been identified recently in breast cancer and several studies already highlignt its potential link with endrocrine-therapy resistance using preclinical and clinical approaches. Consequently the novelty of the data provided by Jae Heon Jeong et al. are limited. Beside ESR1-CCDC170 fusion gene, CCDC170 high expression level seems to be associated with a similar molecular signature. Then comparison of involved molecular pathway and actionable drugs between 3 groups (fusion positive, high expression and low expression) would of major interest to elucidate the real consequence of the fusion.

Authors could have discussed the currently known functions of the CCDC170 protein and the possible interrelationships with the signalling pathways identified in the present study.

Minors concerns:

Line 56, PMID information should be removed.

Line 81: barre code 01 must be explained.

Biocarta website does not work and reference number 16 seems to be a mistake.

Author Response

Response to Reviewer 1 Comments

Point 1: Line 57-59 : Despite suspected, direct and robust clinical evidences of worst prognostic and therapeutic resistance associated with ESR1-CCDC170 fusion gene are still missing. Then authors should be more prudent and avoid misinterpretation of previous work (reference number 13 is exclusively preclinic).

Response 1: We added the term “mouse model” to avoid misinterpretation of previous work on line 67-68 as follows:

“ESR1-CCDC170 fusion-positive cancers treated with endocrine therapy showed reduced treatment efficiency in mouse models [13].”

Point 2: The aim of the study (comparison between CCDC170 low and high expression level or decription of ESR1-CCDC170 fusion gene consequence?) and the selection of control samples are not clear. It seems that there are 4 groups in this study: the whole cohort (named "total" in table 1, whose exact case number remains unclear), the ESR1-CCDC170 fusion gene positive group (n=11), a positive control group displaying high CCDC170 mRNA expression without fusion gene (n=??) and a negative control group displaying low CCDC170 mRNA expression (n= 48 or 44 ??).

Response 2: Thank you for your detailed comment that improving our study. As you concerned, we reviewed our study cohorts and revised the groups. The aim of this study was to analyze the downstream pathways of E:C fusion. The characteristics of E:C fusion were high expression of CCDC170 because the all E:C fusion positive cases were included in the top 50% CCDC170 expression group. Therefore, we compare E:C fusion positive cases (11 samples) to CCDC170 low expression group (48 samples). Additionally, there were various reason of increasing CCDC170 gene expression, for example chromatin remodeling, enhancer hijacking, methylation, and so on, we compared CCDC170 high expression group (48 samples) to CCDC170 low expression group (48 samples) as well. Table 1 was revised to include the nullified samples again. Hence, the final number of samples used in the analysis turned out to be 1,095, 11, 48 and 48, respectively, for following four groups: fusion-positive, high-expression, and low-expression. Clinical data for the high expression without fusion was added to the Supplementary Table 1. In this analysis, we did not consider the CCDC170 high-expression group as a positive control.

Point 3: Selection of control groups (negative and positive) must be explained in details : cut-off to distinguish high and low expression level in comparison with the ESR1-CCDC170 fusion gene positive group, with 50, 48 or 44 cases ? does the total cohort correspond to ESR1-CCDC170 fusion positive group + positive and negative control groups + the remaining cases (normal or medium expression level of CCDC170 protein? Moreover a clear description of clinical characteristics of these 4 groups is expected.  In table 1, we don’t know if the control group correspond to high or low CCDC170 expressing group ?).

Response 3: The reason behind us establishing the cutoff for the number of samples to 50 was in order to sort the expression level of the CCDC170 gene in a descending order. It was confirmed that the increase in the expression level of CCDC170 changed most noticeably near the top 50 samples. Therefore, 50 samples were set as the cutoff limitation point, and 48 samples, the number of removing 2 outliers, were obtained as the high-expression group. As for the 48 high-expression samples without fusion, 48 ​​low-expression samples were selected and set as a control group (Method 2.2 Case Control Selection). The reason for selecting the low-expression group as a control was so that when the fusion group was divided into the top 50% and the bottom 50% according to the CCDC170 expression, all fusion groups were included in the top 50% group. Therefore, the fusion group and the low-expression group (control) were mainly compared, and the high-expression group without fusion was additionally analyzed in the Supplementary table 1 and 3.

Point 4: In addition, exclusion of metastatic cases should be explained (why ?). It seems that they are not fully excluded because in Table 1, stage Iv, which does not exist, is probably stage IV, which is the metastatic stage.

Response 4: Metastatic cases were not excluded, but due to the TCGA cohort characteristics, the number of metastatic cases compared to the total sample was 20 samples, and fusion positive cases and low expression cases were not included.

Point 5: Authors have written about “39 genes identified in Figure 4”, but the list of these genes was not given and must be provided because figure 4 is not contributive to identified them.  

Response 5: Thank you for correcting our errors. The 39 genes were not associated with more than 3 pathways, but 2 pathways. 39 genes list is summarized and added to Table S2.

Point 6: ESR1-CCDC170 fusion gene has been identified recently in breast cancer and several studies already highlight its potential link with endocrine-therapy resistance using preclinical and clinical approaches. Consequently, the novelty of the data provided by Jae Heon Jeong et al. are limited. Beside ESR1-CCDC170 fusion gene, CCDC170 high expression level seems to be associated with a similar molecular signature. Then comparison of involved molecular pathway and actionable drugs between 3 groups (fusion positive, high expression and low expression) would of major interest to elucidate the real consequence of the fusion. Authors could have discussed the currently known functions of the CCDC170 protein and the possible interrelationships with the signalling pathways identified in the present study.

Response 6: Many studies have attempted preclinical and clinical analysis of the ESR1-CCDC170 fusion gene, but no known interaction of CCDC170 has been confirmed in curated databases or experimentally determined data (string database: https://string-db.org/) so far. We conducted further investigations if there is any biological difference between ESR1-CCDC170 fusion-positive group and CCDC170 high expression group without fusion in major cancer related pathways. We found that there showed no major difference in cancer signaling except several minor pathways including cilium assembly pathway and integrins in angiogenesis pathway (Table S3).

We added further explanations on lines 686-689 as follows:

“In addition, we investigated if there is the biological difference between E:C fusion-positive group and CCDC170 high expression group without fusion. We found that there showed no major difference in cancer signaling except several minor pathways including cilium assembly pathway and integrins in angiogenesis pathway (Table S3).”

Minors issues:

  1. Line 56, PMID information should be removed.
  • We removed PMID.
  1. Line 81: barre code 01 must be explained.
  • We explained barcode 01 on line 81 as follows:

“Furthermore, only tumor samples that have the barcode 01A (Primary Solid Tumor) were selectively chosen by disregarding other types of tumor samples, 11A (Normal) or 06A (Metastasized).”

  1. Biocarta website does not work and reference number 16 seems to be a mistake.
  • We replaced the website link on line 101 as follows:

“BioCarta (https://maayanlab.cloud/Harmonizome/dataset/Biocarta+Pathways)”

Reviewer 2 Report

This paper reflects the current problem of precision medicine in breast cancer. It would be good to emphasize this in the introduction. It is very basic scientific knowledge after a data base analysis (TCGA), I miss the clinical correlation of the different approaches, here are some suggestions to improve this: 

  1. Please write in the introduction more about the different resistance mechanisms, ESR1 mutations, RB mutations, loss of receptors, re-biopsy needed, I would like to have a more clinical relationship 
  2. Please explain your data collection a bit more in precession, did you use a publicly available data base as you write, could you explain this to the reader how this will work (please not only a link) 
  3. You stated that 6-8% of all luminal breast cancer have an ESR1-CCDC170 fusion, in your analysis you presented 1000 cases, why only 10 cases with ESR1-CCDC170 
  4. Any idea if a SERD will work better in patients with ESR1-CCDC170 fusion? 
  5. Why did you choose ONKOKB and the other data bases, what is the scientific strength of this?

For a clinician it sounds good to have a new targeted therapy. please outline if there is any drug on the horizon that might target the fusion directly 

Author Response

Response to Reviewer 2 Comments

Point 1: Please write in the introduction more about the different resistance mechanisms, ESR1 mutations, RB mutations, loss of receptors, re-biopsy needed, I would like to have a more clinical relationship 

Response 1: We added the different resistance mechanisms to emphasize the importance of drug repositioning analysis in discussion sections on lines 665- 670 as follows:

“On the other hands, many resistance mechanisms for drug therapy in breast cancer have been reported as follows: loss of estrogen receptor, deregulation of cell cycle for endocrine therapy; incomplete blockade of HER receptors, activation of the PI3K pathway, and overexpression of estrogen receptor for HER2 inhibitors; polyclonal RB1 mutations for CDK 4/6 inhibitors and so on [25,26]. Hence, finding novel therapeutic strategies using drug repositioning analysis is crucial for modern breast cancer treatment.”

Point 2: Please explain your data collection a bit more in precession, did you use a publicly available data base as you write, could you explain this to the reader how this will work (please not only a link) 

Response 2: We added the explanations and conditions for the public database used on lines 121-124 as follows:

“We input gene list with option of entrez gene using pathway-based sets with minimum overlap input list (n=2) and p-value cutoff (p-value < 0.01). 113 biological pathways were merged and curated by CPDB from the following sources, according to data from BioCarta (https://maayanlab.cloud/Harmonizome/dataset/Biocarta+Pathways), INOH [16], KEGG [17], NetPath [18], PID [19], Reactome [20] and Wikipathways [21].”

Point 3: You stated that 6-8% of all luminal breast cancer have an ESR1-CCDC170 fusion, in your analysis you presented 1000 cases, why only 10 cases with ESR1-CCDC170 

Response 3: Referring to Table 1, there are 783 luminal cases, of which 6-8% of them should be calculated as 46-62 cases of CCDC170 fusion positive samples. However, there are three main reasons why only 11 fusion-positive cases were found in the entire sample. First, the characteristics of the TCGA cohort and the existing reported cohort may be different basically. Second, we obtained the data from the Jackson lab which may pose a problem with regard to the sensitivity of the fusion detection algorithm (PRADA) in RNA-sequence. Finally, tumor heterogeneity and tumor purity may have lowered the fusion detection rate. For these reasons, the number of samples with E:C fusion estimated numerically and the actual number of samples with E:C fusion from Jackson lab may be different.

Point 4: Any idea if a SERD will work better in patients with ESR1-CCDC170 fusion? 

Response 4: ESR1 was upregulated in CCDC170 fusion (figure 5), and it was predicted that the upregulated ESR1 may be regulated by fulvestrant (SERD). However, additional clinical validation is required via in vitro and in vivo studies.

Point 5: Why did you choose ONCOKB and the other data bases, what is the scientific strength of this?

Response 5: OncoKB database is from MSKCC (Memorial Sloan Kettering Cancer Center) and CIViC database is from Washington University. As such, these are all databases curated by experts, and in our experience, when comparing the above two databases with experimental studies and literature reviews, the results were more than 95% accurate. However, I think that data of pre-clinical level (eg. level C or level D) may need additional verification.

Point 6: For a clinician it sounds good to have a new targeted therapy. please outline if there is any drug on the horizon that might target the fusion directly 

Response 6: The drugs we presented are putative drugs that can be repositioned through the downstream pathway analysis of E:C fusion. On the other hand, I think that the additional drug discovery and experimental validation may also be needed to discern drugs that directly target fusion.

Round 2

Reviewer 2 Report

Thank you for integrating my comments